

# Development and performance of a new version of the OASIS coupler, OASIS3-MCT_3.0

Anthony Craig[1], Sophie Valcke[1], Laure Coquart[1]

[1]CERFACS, Sciences de l'Univers au CERFACS, URA1875, 42 Av. G. Coriolis, 31057 Toulouse Cedex 01, France

*Correspondence to*: S. Valcke (valcke@cerfacs.fr)

**Abstract.** OASIS is coupling software developed primarily for use in the climate community. It provides the ability to couple different models[1] with low implementation and performance overhead. OASIS3-MCT is the latest version of OASIS. It includes several improvements compared to OASIS3 including elimination of a separate hub coupler, full parallellisation of the coupling communication and grid interpolation, and the ability to easily reuse mapping weights files. OASIS3-MCT_3.0 is the latest release and includes the ability to couple between components running sequentially on the same set of tasks as well as to couple within a single component between different grids or decompositions such as physics, dynamics, and I/O. OASIS3-MCT has been tested with different configurations on up to 32,000 processes, with components running on high-resolution grids with up to 1.5 million grid cells, and with over 10,000 two dimensional coupling fields.

## 1 Introduction

OASIS is coupling software developed primarily for the climate community. It is in relatively wide use especially in European based modeling efforts. It is one of a number of coupling infrastructure packages (Valcke et al., 2016) that is focused on standard and reusable methods to support coupling requirements like interpolation and communication of data between different models and different grids. OASIS is maintained and managed by the Centre Européen de Recherche et de Formation Avancée en Calcul Scientifique (CERFACS) and the Centre National de la Recherche Scientifique (CNRS) in France. It is a portable set of Fortran 77, Fortran 90 and C routines. Low-intrusiveness, portability and flexibility are key OASIS design concepts. The current version of the software, OASIS3-MCT, is a coupling library that is compiled and linked to the component models. Its primary purpose is to interpolate and exchange the coupling fields between or within components to form a coupled system. OASIS3-MCT supports coupling of fields on relatively arbitrary grids via a put/get approach. This approach means components make subroutine calls to send (put) or receive (get) data from within the component code directly. A top-level driver is not required to use OASIS3-MCT, but a handful of subroutine calls must be added to the code to

---

[1] Within the text, we use "model" in the sense of a "numerical model"



initialize the coupling, define grids, define decompositions (partitions), define coupling fields, and to put and get variables between components. OASIS3-MCT leverages a text input file called the namcouple file to configure the interactions between components. Mapping (also known as remapping, regridding, or interpolation), time transformations, and the ability to read or write coupling data from disk are supported

in OASIS3-MCT.

OASIS development began in 1991 and the first version, OASIS1, was used two years later in a 10-year coupled integration of the tropical Pacific (Terray et al., 1995). In the intervening decades, OASIS2 and OASIS3 were released. The history of OASIS development is well documented (Valcke, 2013). With

OASIS3, the coupled models always had to run concurrently as separate executables on different MPI tasks and all coupling fields passed through a separate hub coupler that also ran concurrently. OASIS3 allowed parallel coupling of parallel models only on a per-field basis. Each parallel field in the source model was gathered to a single process on the hub where operations such as mapping and time averaging were executed, and the field was then scattered to the destination model. OASIS3 generated mapping weights on

a single process at initialization using the SCRIP library (Jones, 1999) from the grid information specified by the component models.

A first attempt to design and develop a fully parallel coupler was started in the framework of the EU FP5 PRISM and FP7 IS-ENES projects (see https://is.enes.org, PHASE 1) and led to the development of

OASIS4 (Redler et al. 2010). In particular, OASIS4 included a library performing a fully parallel calculation of the source neighbour weights and addresses needed for the mapping of the coupling fields. This version had several other features such as the use of an xml file for specifying the configuration information. OASIS4 was used by Météo-France, ECMWF, KNMI and MPI-M in the framework of the EU GEMS project for 3-D coupling between atmospheric dynamic and atmospheric chemistry models, and by

SMHI, the Alfred Wegener Institute for Polar and Marine Research (AWI in Germany), and by the Bureau of Meteorology in Australia for ocean-atmosphere 2-D regional and global coupling. But OASIS4 had limited success and its development was stopped in 2011 after a performance analysis determined some fundamental weaknesses in its design, in particular with respect to the support of unstructured grids.

With OASIS3-MCT, a different approach was taken to improve the parallel performance and to address

new requirements. OASIS3-MCT extended the widely used and distributed OASIS3 version of the model. The initial requirements of OASIS3-MCT were to improve the parallel performance of the coupling, implement an ability to read in mapping weights to improve reuse and mitigate the cost of weights generation, support next generation grids such as high resolution unstructured grids running on high processor counts, and to add those features while retaining the basic OASIS3 application programming

interfaces (APIs) and namcouple file to support backwards compatibility.



To accomplish these requirements, a number of changes were made. First, a portion of the underlying communication implementation was replaced with the Model Coupling Toolkit (MCT) software package (Larson et al., 2005) developed by the Argonne National Laboratory. This implementation is transparent to the user, as MCT methods and datatypes are only used within the OASIS3-MCT infrastructure to support

parallel mapping and parallel redistribution. Second, the ability to specify pre-defined mapping files was added. Mapping files can now be generated offline using a diverse set of packages, such as SCRIP, ESMF (Theurich et al, 2016), or any locally developed methods. Third, the hub coupler was deprecated and is no longer required. Transforms are carried out on the source or destination processes, and data is transferred directly between components via MCT. These features were released in OASIS3-MCT_1.0 in 2012

(Valcke et al, 2012b) and because of the backwards compatibility, OASIS3 users could upgrade easily to OASIS3-MCT.

With the release of OASIS3-MCT_3.0 in 2015, several new features were added to the coupler. OASIS3-MCT_3.0 extends the ability to couple components running concurrently and adds support for coupling

components running on overlapping or partially overlapping sets of tasks, even within a component, such as between physics and dynamics modules within an atmospheric model or to and from a model I/O module. OASIS3-MCT_3.0 also allows a component to define grids, partitions, and coupling fields on subsets of its tasks and comes with a Graphical User Interface (GUI) to generate the namcouple file.

The next section, titled implementation, describes these features in greater detail. Section 3 provides performance and memory scaling results from OASIS3-MCT_3.0, and section 4 provides a summary.

## 2. Implementation

As discussed in the introduction, OASIS3-MCT development started with the objective to keep the

OASIS3 general design. The requirements of OASIS3-MCT were focused on improved parallel performance including parallel mapping and parallel data coupling, the ability to efficiently support unstructured grids, an ability to specify pre-defined mapping files to mitigate the serial cost of generating mapping weights on-the-fly, and backwards compatibility in usage of both the namcouple file and the OASIS3 APIs. A summary of the changes between OASIS3 and OASIS3-MCT_3.0 is provided in

Appendix A.

### 2.1 General Architecture

To accomplish these tasks efficiently and in a timely manner, the Model Coupling Toolkit (MCT) developed by the Argonne National Laboratories (Larson et al 2005) was incorporated into OASIS3 to

support parallel matrix vector multiplication and parallel distributed exchanges. Its design philosophy,



based on flexibility and minimal invasiveness is consistent with the approach taken in OASIS. MCT has proven parallel performance and is one of the underlying coupling software libraries used in the National Center for Atmospheric Research Community Earth System Model (NCAR CESM) (Jacob et al., 2005, Craig et al., 2012).

MCT handles two primary tasks in OASIS3-MCT. The parallel transfer of data from a source model to a destination model, and interpolation of fields between decomposed grids. At the present time, these two steps are independent and both are largely performance limited by MPI communication cost at moderate to high processor counts due to the data rearrangement in both. Data coupling and mapping communication is
handled internally in OASIS3-MCT via MCT routers.

Another significant change in the OASIS3-MCT implementation compared to OASIS3 is that a separate coupler executable running on its own processes is no longer needed. Accumulation, temporal lagging, mapping, and other transforms are carried out in the OASIS3-MCT coupling layer on the model processes
in parallel using temporary memory to store data as needed. Compared to OASIS3 which required two data rearranges to couple fields in order to pass through the hub, OASIS3-MCT requires just one parallel rearrange to move data between two components. In addition, the memory needed in the infrastructure is much more scalable.

### 2.2 Coupling

OASIS3-MCT fundamentally supports coupling of 2-D logically rectangular fields but 3-D fields and 1-D fields are also supported using a one-dimension degeneration of the grid structure. If the user provides a set of pre-calculated weights, OASIS3-MCT will be able to interpolate any type of 1-D, 2-D, or 3-D fields, but the capability to calculate the mapping weights by the coupler is only available for 2-D fields on the sphere.

Another new feature is the option to couple multiple fields as a single coupling operation. This is supported for fields for which the coupling options defined in the namcouple file are identical. This can improve performance because rather than mapping and coupling fields one at a time, the mapping and coupling can be aggregated over multiple fields. Coupling multiple fields at once is accomplished by
specifying a list of colon-delimited fields in the namcouple file on both the source and destination side. In this implementation, the get and put calls in the model are still individual calls on individual fields, but the coupling layer will aggregate the multiple fields specified in the namcouple file into a single step. On the put side, the multiple fields are not mapped or sent until all of the individual put calls are made. On the get side, the multiple fields are received and mapped on the first get call and then subsequent get calls just copy
in fields that were received earlier. A user can quickly switch between coupling single and multiple fields just by changing the namcouple input file.



One additional feature available in the current development version and that will be released with the next official version, OASIS3-MCT_4.0, is the ability to couple a bundle of 2-D fields via extensions to the OASIS calling interfaces. An extra dimension is supported in the variable definition and in the get and put field arrays. In this case, a user can treat a bundled 2-D field as a single field in the system, while the underlying implementation treats it just like a multiple field coupling.

### 2.3 Interpolation

Mapping weight files can either be read directly or they can be generated at run-time, on one processor, using the same serial method based on SCRIP as existed in OASIS3. For the interpolation, OASIS3-MCT creates a simple one-dimensional decomposition of the source grid on the destination processes or vice-versa. Fields are then either interpolated on the source processes and sent to the destination processes or sent to the destination processes and then interpolated to the destination grid. The user is able to specify whether the source or destination processes are used for mapping via an optional setting in the namcouple file. That choice will generally be made based on mapping performance and depends on the relative size of the grids, the number of weights, and the process counts of the source and destination models.

Users also have an additional option to specify the type of mapping to be carried out. The *bfb* option will enforce an order of operations that will be bit-for-bit identical on different process counts. It does this by distributing the mapping weights on the destination decomposition and then redistributing the source coupling field grid point values to the destination processes before applying the mapping weights. This ensures operation order is independent of decomposition. The *sum* option does the opposite. It distributes the mapping weights on the source decomposition and then computes partial sums of the destination field on the source decomposition, before redistributing them to the destination decomposition and adding up the partial sums. This does not guarantee identical order of operations on different processor counts and decompositions. In both approaches, the same number of floating operations are carried out; those are defined by the mapping weights. The main difference between the *bfb* and *sum* strategies is that in *bfb* mode, the source field is rearranged onto the target distribution before the mapping weights are applied while in *sum* mode, the mapping weights are applied on the source decomposition to form partial sums of the destination field and then the partial sums are rearranged. From the performance point of view, it's generally better to use the method that rearranges the grid that contains the fewest grid cells to minimize the communication cost. But of course, if bit-for-bit reproducibility on different core counts is required, then the *bfb* mode should be chosen.

### 2.4 Conservation



With OASIS3-MCT, the CONSERV transform has been refactored. In OASIS3, this operation was always performed on a single process. In OASIS3-MCT, this operation is now performed in parallel on the source or destination processes and is an inherent part of the mapping operation. The CONSERV operation computes global sums of the source and destination fields and applies corrections to the mapped field in order to conserve area-integrated field quantities. There are currently two options for computing the global sums in OASIS3-MCT. The first, *bfb*, gathers the fields onto the root process to compute the global sums in an ordered fashion that guarantees bit-for-bit identical results regardless on the number of cores or decomposition of the field. The second option, *opt*, carries out a local sum of the field and then does a scalar reduction to generate the global sums. This will typically introduce a round off difference in the results when changing process counts but is much faster. The *opt* option will however be bit-for-bit reproducible if the same number of processes is used between different runs.

### 2.5 Concurrency, Process Layout, and Sequencing

The ability to couple fields within one executable between components running on partially overlapping tasks was added in OASIS3-MCT_3.0. A number of new capabilities had to be implemented to support this feature including the ability to define grids, partitions, and coupling fields on subsets of component tasks. There also had to be a major update in the handling of MPI communicators within the infrastructure. These changes are transparent to the user. This allows, within a single model, different sets of MPI tasks to define multiple grids, multiple decompositions (partitions), and different coupling fields. These new features and updates provide the flexibility needed to couple fields between components or within a component.

Figure 1 provides a schematic of the type of coupling that can be carried out between and within components in OASIS3-MCT_3.0. In OASIS3-MCT_3.0, executables are defined as separate binaries that are launched independently at startup, components are defined as separate sets of tasks within an executable, and grids can be defined on all tasks or on a subset of tasks within a component. Each task will be associated with only one executable and one component in any application, but multiple grids and decompositions can exist across overlapping tasks within a component.

In figure 1, an example schematic is presented that shows how two executables, exe1 and exe2, are running concurrently on separate sets of MPI tasks (0-5 for exe1 and 6-37 for exe2). Executable exe1 includes only one component comp1 that has coupling fields defined on only one grid, grid1 (decomposed on all 6 tasks). Executable exe2 includes 3 components, comp2, comp3, and comp4 running concurrently on tasks 6-11, 12-33 and 34-37 respectively. Component comp2 participates in the coupling with fields defined on only one grid, grid2 (decomposed on all 5 tasks) while comp4 does not participate in the coupling. Component comp3 exchanges coupling fields defined on 3 different grids, grid3 (tasks 12-21), grid4 (tasks 22-30) and grid5 (tasks 12-26, overlapping with both grid3 and grid4). Finally, comp3 has 3 tasks (31-33) not



involved in the coupling. Different coupling capabilities are indicated by the different lettered arrows in Figure 1. Coupling is supported between components in separate executables, within a single executable between different components, and between overlapping, non overlapping, or partially overlapping grids in a single component. In OASIS3, only coupling between separate executables was supported; in OASIS3-

MCT_3.0, a functional and highly flexible coupled system can now be designed and implemented as either a single executable or with multiple executables.

Figure 2 shows how the OASIS3-MCT_3.0 API calls should be executed across different tasks to support the coupling shown in figure 1. Each MPI tasks has to call the oasis initialization routine (oasis_init_comp)

once with the name of its component. Comp4 is not participating in coupling, so that component calls the oasis initialization routine with the argument coupled=.false. and then that component does not need to call any other OASIS3-MCT routine. Since some of comp3 tasks are participating in the coupling, all comp3 tasks have to call the routines to initialize the coupling (oasis_init_comp), retrieve a local communicator for all component processes (oasis_get_localcomm), create a coupling communicator

(oasis_create_couplcomm), finalize the definition phase (oasis_enddef), and terminate the coupling (oasis_terminate), and these are the only routines that have to be called by comp3 tasks 31-33 since those tasks are not participating in the coupling. To initialize the coupling exchanges, the tasks that participate in coupling have to define the decomposition of the grids (oasis_def_partition) and declare the coupling fields (oasis_def_var). Finally, the tasks exchanging coupling fields have to call the sending (oasis_put) and

receiving (oasis_get) routines accordingly.

Within OASIS, it has always been mandatory for a user to establish a set of configuration inputs that are consistent with the get and put sequencing in the components such that the coupled system will not deadlock. OASIS3-MCT provides some new capabilities to detect potential deadlocks before they occur,

but it is still largely up to the user to make sure this does not happen. This is even more important for coupling components on overlapping tasks as there is almost no way to detect a deadlock ahead of time. Specifically, a field put routine must be called before the matching get (taking into account any lags specified in the configuration file) when coupling on overlapping tasks. In OASIS3-MCT, puts are always non-blocking while gets are blocking.

**2.6 Other Features**

There are several additional features in OASIS3-MCT relative to OASIS3. The grid writing routines have been extended to support parallel calls from all component processes. However, even when the parallel interface is used, the grid information is still aggregated onto the root processor within the OASIS3-MCT

layer and then written serially to disk.





OASIS3-MCT also now includes a GUI, which is an application of OPENTEA (Dauptain, 2014), the graphical interface developed at CERFACS. The OASIS3-MCT GUI helps users produce the namcouple configuration file for a specific run, without worrying about the format syntax of the file.

### 3. Performance

This section summarizes the performance of various aspects of OASIS3-MCT_3.0 at low and high process counts and at moderate to high resolution. The performance and scaling of initialization, coupling, mapping, conservation and other features will be presented. Memory usage will also be shown.

### 3.1 Initialization

Figure 3 shows the initialization cost for a T799-ORCA025 test case on up to 16,000 cores per component running concurrently (32,000 tasks total) on Curie at CEA's TGCC. Curie consists of 5040 nodes with 16 Intel Sandy Bridge 2.7GHz processes per node connected with an InfiniBand QDR Full Fat Tree network. These tests were run with simple toy models that define grids, couple test data, but have practically no model initialization or run-time overhead. This configuration was chosen because it demonstrates OASIS3-MCT's ability to support high-resolution climate configurations. The T799 is a global atmospheric gaussian reduced grid with a ~25km resolution and 843,490 grid points. The ORCA025 grid is a tripolar grid with 1442 x 1021 (~1.47 million) grid points and is based on a grid from the NEMO ocean model (http://www.nemo-ocean.eu/). The OASIS3-MCT initialization consists of several steps including setting up the partitions, reading in and distributing the mapping weights, computing the mapping communication patterns, and computing the coupling communication patterns. Most of these operations rely heavily on MPI to define the interactions, reconcile the coupling fields and decompositions, and setup the mapping and coupling interactions. Multiple runs were performed for each number of cores with little variability in timing measured. Based on the results in figure 3, the total initialization time for Oasis3-MCT is likely to be reasonable for most applications, even at high numbers of cores. Below 2000 cores (MPI tasks) per component, the OASIS3-MCT initialization time is less than one minute. At 16,000 cores per component, for this relatively high-resolution configuration, the initialization time is below 7 minutes. There is however clearly some concern that as core counts continue to increase, the initialization time will continue to grow. But one has to consider that in many ways, the time spent setting up complex MPI interactions during the initialization is what allows the model to run efficiently during the actual coupling run phase.

### 3.2 Coupling





Figure 4 shows the cost of a ping-pong communication for the same configuration as figure 3. The timings are per single ping-pong communication but the test was done by running and averaging 1000 ping-pongs. In a ping-pong test, data is passed back and forth between the two components sequentially. In other words, data is sent from model 1 and received by model 2, followed by different data being sent from model 2 to

model 1. Each coupling of data between a pair of components consists of a mapping operation that interpolates the data via a two-nearest-neighbor algorithm that includes both floating point operations and rearrangement, and then a redistribution operation that transfers the data between concurrent sets of MPI tasks in the different components. So there are four distinct MPI operations in a single ping-pong. There are 4.5 million different links (weights) between the T799 grid points and the ORCA025 grid points, and

there are 3 million weights for the mapping in the other direction. In this case, scaling is good to about 400 cores per component as the MPI cost is relatively small and the floating point operations associated with the mapping dominate the cost. Between 400 and 4000 cores per component, the ping-pong cost is relatively constant and above 8000 cores per component, the timing is degraded relative to lower core counts. At higher core counts, the timing depends heavily on the MPI performance. At 8000 cores per

component, decompositions are getting relatively sparse with just 100 to 200 grid points per core. In addition, timing variability between runs (not shown) above 1000 cores and the jump in cost at 8000 cores suggests that interconnect contention is likely a problem at these core counts. Equivalent timings from OASIS3.3 are also shown in Figure 4 (Valcke, 2013), and the ping-pong time is about an order of magnitude better in OASIS3-MCT for a large range of the core count.

**3.3 Interpolation**

One of the features of OASIS3-MCT is the ability to map data on either the source or destination side as described in Section 2.3. Figure 5 shows the timing of the mapping portion of coupling which includes both the floating point application of weights and the necessary rearrangement of the data on either the

source processes (*src*) or the destination processes (*dst*) but not the communication between the source and target process. This test was run using the T799-ORCA025 toy model on a Lenovo Xeon based cluster at CERFACS consisting of over 6000 2.5 GHz cores connected by an Infiniband FDR. Mapping is about half the total cost of the ping-pong (not shown) in these cases. Figure 5 shows timing data for both mapping directions and for mapping done on the source (*src*) or destination (*dst*) side. In all cases, the *bfb* algorithm

is used. The mapping in this case scales well to several hundred cores. In general, the cost of the T799 to ORCA025 mapping is more expensive than the reverse, largely due to the fact that there are more mapping weights (4.5 vs 3.0 million) to apply.

Table 1 documents the ping-pong time for 1000 trials for the same T799-ORCA025 toy model test on

Lenovo. In this case, the total number of cores is held at 360, but the relative distribution of cores to each model is varied in three test configurations. In addition, the ping-pong tests were carried out with the



mapping done on the source, the destination, the ORCA025, or the T799 sets of cores. In these trials, the *bfb* map algorithm was used. The best performance is when the mapping is done on the model with the highest core count because in this range of core counts, the mapping and communication are still scaling. At higher core counts, the optimum performance may be different. For the current cases, the best time is a

5 factor of up to 2.5 times better (1.91 vs 4.70) compared to the default setting of *src* and by an even greater factor compared to the slowest setting. In general, the number of processes per component is not going to be determined by the coupling cost but by the relative cost of the scientific models, but the above analysis shows that for a given task layout, there may be ways to reduce the coupling cost by mapping on the tasks that provide the greatest performance.

**3.4 Field Aggregation**

OASIS3-MCT provides a new feature, as described in Section 2.2, that allows users to aggregate coupling of multiple fields into a single coupling operation by specifying coupled fields via colon delimited field names in the namcouple file. Tables 2 and 3 show mapping and ping-pong timing for the T799-ORCA025

configuration on Lenovo using single and multiple fields. Table 2 shows the mapping time on 180 cores per component for a single field, 10 fields coupled via 10 coupling calls, 10 fields coupled via a single coupling communication, and 10 fields bundled into a single variable. In general, the time to couple 10 fields is proportionally less than the time to couple 1 field, and a clear advantage is seen in the 10 field mapping cost when done as a single aggregated operation. The time for mapping a bundle field of

dimension 10 is similar to the time for the 10 fields coupled via a single coupling; this is expected because the underlying implementation is basically the same.

Table 3 shows the ping-pong times for the same cases as table 2. In this case, the advantage of aggregating multiple coupling fields is less clear. For the *dst+bfb* case, the single operation performs only marginally

better while for the *src+bfb* case, the single operation performs slightly worse. In this case, the coupling calls are happening sequentially without any real model work between calls. In a real model, the coupling performance will depend on the sequence of the coupling calls, how much work can be overlapped with the coupling, and the relative core counts and grid sizes of the different coupling fields.

**3.5 Conservation**

Table 4 shows the timings of a ping-pong test of the T799-ORCA025 case on the Lenovo cluster for four different configurations and cases with CONSERV unset, CONSERV set to *opt*, and CONSERV set to *bfb*. The CONSERV implementation and its *bfb* and *opt* options are described in Section 2.4. Table 4 shows that the CONSERV, even when the global sum option is set to *opt*, adds a cost to the coupling. However,

beyond that, the difference in cost between the *opt* and *bfb* CONSERV settings are significant for this high





resolution case because the *bfb* option gathers the entire field on the root process while the *opt* routine uses a more parallel algorithm and only gathers a scalar from each task. When using CONSERV, it is therefore important to test the performance of both *opt* and *bfb* and decide whether it's absolutely necessary to use *bfb*. Of course, when possible, mapping weights that are inherently conservative (such as area overlap conservative) should be used to avoid use of the CONSERV operation all together.

### 3.6 Memory

Figure 6 shows the memory use per core for the T799-ORCA025 test case on Curie, the same test case as Figures 3 and 4. Memory use was determined by calls into the gptl (http://jmrosinski.github.io/GPTL/) interface which queries memory usage through C intrinsics and is included in the OASIS3-MCT release. At 16,000 cores, the infrastructure is using a bit more than 1GB per core which, while not tiny, is generally acceptable for many applications and hardware. Memory is increasing on a per core basis at higher core counts. It's likely that the MPI memory footprint is accounting for most of this behavior (Balaji et al, 2008, Gropp, 2009). The fundamental datatypes and arrays in OASIS3-MCT, such as fields and partitions, are generally quite memory scalable as implemented, and we do not believe the memory increase with core count is coming primarily from OASIS3-MCT.

### 4. Conclusions

OASIS3-MCT was implemented largely to address limitations in parallel performance of OASIS3 and to provide a framework for use at higher resolutions. With OASIS3-MCT, the widely used OASIS3 model interfaces (APIs) and configuration file have largely been preserved, and this explains the wide adoption of OASIS3-MCT within the OASIS user community. Since its release in May 2015, about 250 downloads of OASIS3-MCT_3.0 were registered from most major climate modeling groups in Europe as well as from groups in North and South America, Asia, Australia, and Africa. In the last two years, the OASIS3-MCT coupler was used in many state-of-the-art coupled systems including high resolution climate models and systems that couple 3-D atmospheric fields between global and regional models frequently among others. Other examples of coupled model applications that use OASIS3-MCT can be found on the OASIS3-MCT coupled model page[2].

The underlying software was refactored significantly in OASIS3-MCT to improve parallel performance and coupling capabilities. MCT serves as a key part of the OASIS3-MCT implementation and provides parallel capabilities for coupling operations. OASIS3-MCT_3.0 also provides new capabilities to couple fields within a single component or between components running on overlapping or partially overlapping

---

[2] https://verc.enes.org/oasis/oasis-dedicated-user-support-1/some-current-coupled_-models/some-oasis3-mct-coupled-models



processes. This increases the flexibility of OASIS3-MCT significantly and provides a mechanism for coupling data between different decompositions or grids within a single model among many other things. OASIS3-MCT can now be used as a coupling layer for components running sequentially, concurrently or both; for single or multiple executable execution; to exchange coupling fields defined on a subset of the

component tasks ; and to support features like a separate I/O component included in the executable but not involved in the coupling. This provides significant flexibility to layout models on parallel tasks in relatively arbitrary ways to optimize overall performance and to build new features into a model beyond model coupling. OASIS3-MCT has been tested at high resolution, at high processor counts, and with a large number of coupling fields successfully.

There are other benefits in the OASIS3-MCT implementation. Mapping files can be read in directly, more easily reused, and the cost associated with generating the mapping files can be moved to a preprocessing step, but OASIS3-MCT still supports mapping weights generation on-the-fly via SCRIP using a single processor just like OASIS3. There are new features that support creating grid data using a parallel interface, that couple multiple fields in a single operation, and that generate the namcouple file offline via a GUI.

The requirement for an OASIS3 hub coupler has been removed and all coupling and mapping is done in parallel and performance is significantly improved.

The scaling and performance results in Section 3 demonstrate the ability of OASIS3-MCT to support high-resolution model coupling on large core counts. However, as core counts get well into the 10s of thousands

and beyond, there are questions and concerns about the cost of both the initialization and coupling exchanges in OASIS3-MCT. The operations in OASIS3-MCT are ultimately constrained by MPI performance at those core counts, and developers will need to continue to pursue performance improvements in the underlying implementation. However, for the near term future, say the next 5 years, OASIS3-MCT is likely to adequately meet the needs of the climate modeling community.

The flexibility and relative cost of OASIS3-MCT to map fields by various approaches was shown. A general recommendation is to test different approaches and to choose the approach that yields the fastest performance. While it is always first recommended to use conservative mapping weights to avoid the use of the global CONSERV transformation, the performance of the different options of this transformation

were shown for a high-resolution case. If the CONSERV transformation is needed, the more efficient *opt* option, implemented using partial sums, is recommended unless bit-for-bit reproducible results on different core counts are absolutely required. The partial sum option will nevertheless produce bit-for-bit reproducible results for a configuration with fixed process counts and will introduce no more than roundoff level differences when changing process counts. An update of the CONSERV implementations is planned

in the future to improve performance.



The ability to couple multiple fields via a single coupling operation was demonstrated. While not shown in this study, OASIS3-MCT has been used to successfully couple over 10000 fields in some coupled systems within the community. Those tests were carried out both with single field coupling and multiple field coupling with success. In that case, multiple field coupling significantly reduces the size of the namcouple file. Multiple field coupling was shown to reduce the mapping time compared to coupling the same number of fields individually. The performance benefit of using the multiple field feature in the overall coupling time is less clear and will depend on the sequencing and design of each coupled system.

A number of future extensions are being considered for OASIS3-MCT. In theory, it should be possible to combine the mapping and coupling steps to eliminate a field rearrangement and further reduce communication cost. If that is technically difficult, there are several decomposition strategies that could be introduced to reduce the rearrangement cost in the mapping operation. The performance and memory scaling of initialization and coupling may become a larger problem as resolutions and core counts continue to grow. There are also many opportunities in OASIS3-MCT to improve the I/O performance. In the current version, I/O is done via a gather and/or scatter to/from a root task and data is written in serial from the root task. This is likely to eventually lead to memory and performance issues. Finally, better support within OASIS3-MCT for shared memory threading (i.e. OpenMP) and on various multi-core architectures is likely to become more important in the future.

In summary, OASIS3-MCT_3.0 is the latest version of the OASIS coupler. It extends the well-used OASIS software with backwards compatibility with regard to usage, but has an entirely new implementation internally. It provides the functional capability to couple high resolution structured or unstructured grids at high core counts successfully and should serve the community well for the next several years.

**Code Availability**

The OASIS3-MCT source code is available for use and testing after registration at https://verc.enes.org/oasis/download. The SVN command line to get the OASIS3-MCT_3.0 sources is "svn checkout https://oasis3mct.cerfacs.fr/svn/branches/OASIS3-MCT_3.0_branch/oasis3-mct". The OASIS3-MCT_3.0 sources are also available as a tar file at ftp://ftp.cerfacs.fr/pub/globc/exchanges/distrib-oasis/oasis3-mct.tar.gz.

**Appendix A**

The following list provides a history of changes to OASIS3-MCT since OASIS3 and up to OASIS3-MCT_3.0.





OASIS3-MCT_1.0 (2012):

- requirement for separate coupler processes and hub removed
- use of MCT in underlying coupling layer for regridding and communication

- parallel remapping
- fully parallel communication
- ability to couple a single field to multiple destinations
- extended ability to read mapping file
- improved deadlock trapping

- only MPI1 job launching supported
- ability to couple on a subset of processes
- support for one-dimensional coupling field arrays
- support for prism_ and oasis_ interface names
- restart files for LOCTRANS operations

- coupling multiple fields through a single namcouple entry

OASIS3-MCT_2.0 (2013)

- support for bicubic interpolation given the field gradient is specified in the interface arguments
- coupling support on a subdomain of the full grid

- update to timing and debugging capabilities
- parallel interface to grid writing

OASIS3-MCT_3.0 (2015)

- improved memory use, initialization cost and scaling

- updated mapping file reading algorithm
- ability to implement a coupled system within a single executable
- ability to couple sequentially and on partially or completely overlapping processes

**Acknowledgements**

This research was supported by the ESiWACE H2020 European project grant agreement No 675191
(www.esiwace.eu), the IS-ENES2 FP7 European project contract number 312979
(https://verc.enes.org/ISENES2), and the CONVERGENCE project funded by the French National
Research Agency ANR-13-MONU-0008.



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





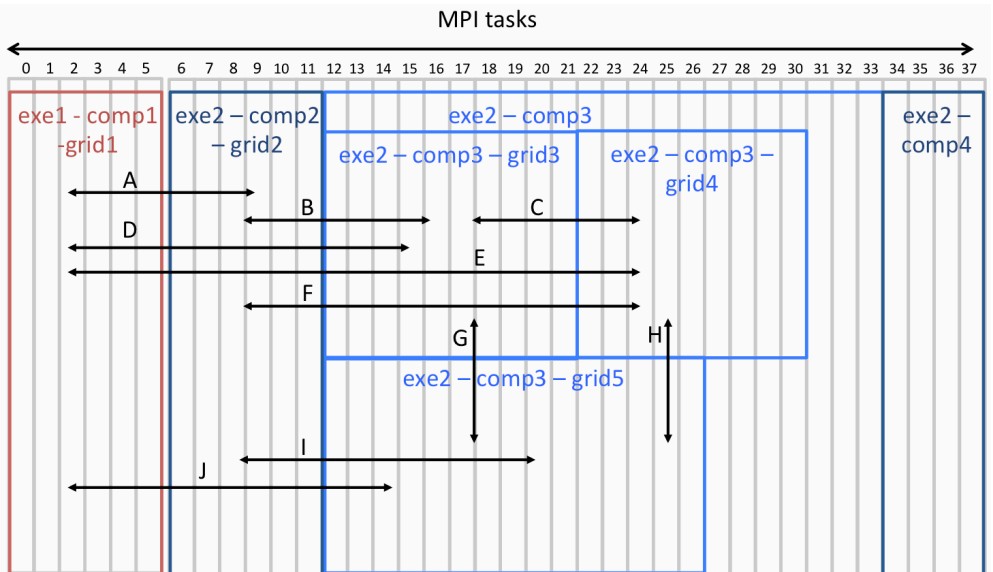

Figure 1. A schematic of the coupling capability in OASIS3-MCT_ 3.0. In this example, there are 2 executables, exe1 and exe2. Executable 2 has 3 components, comp2, comp3, and comp4, and comp3 has 3 grids, grid3, grid4, and grid5; comp4 is not involved in any coupling in this case. The executables, components, and grids are laid out across different tasks. Arrows indicate different coupling capabilities; A), D), E), and J) between different components in different executables; B), F), and I) in a single executable between different components with different grids; C) between different grids in a single component on non-overlapping tasks; G) between different grids in a single component on partially overlapping tasks; and H) between different grids in a single component on partially overlapping and partially non overlapping tasks.





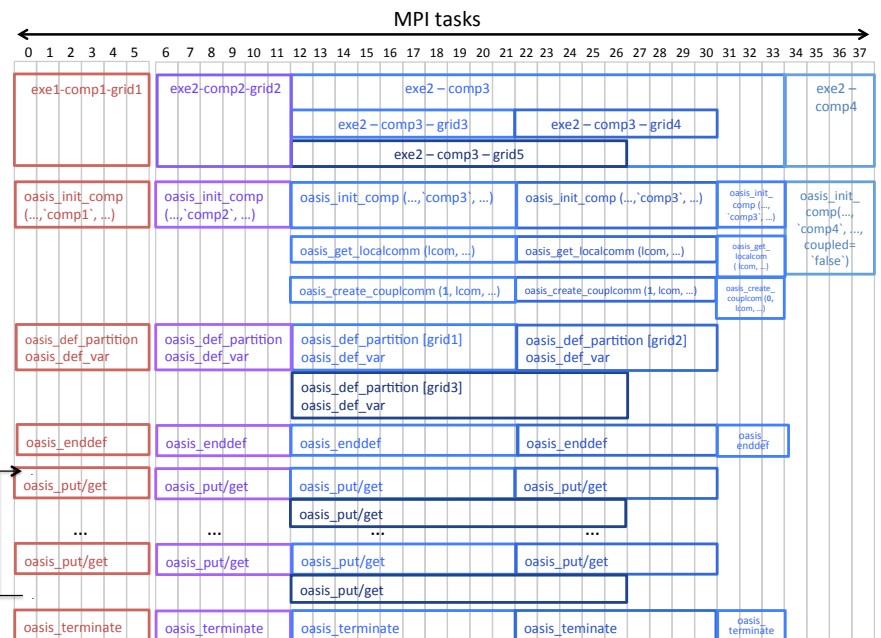

**Figure 2. Schematic of OASIS3-MCT_3.0 coupling calls required to implement the coupling shown in Figure 1.**




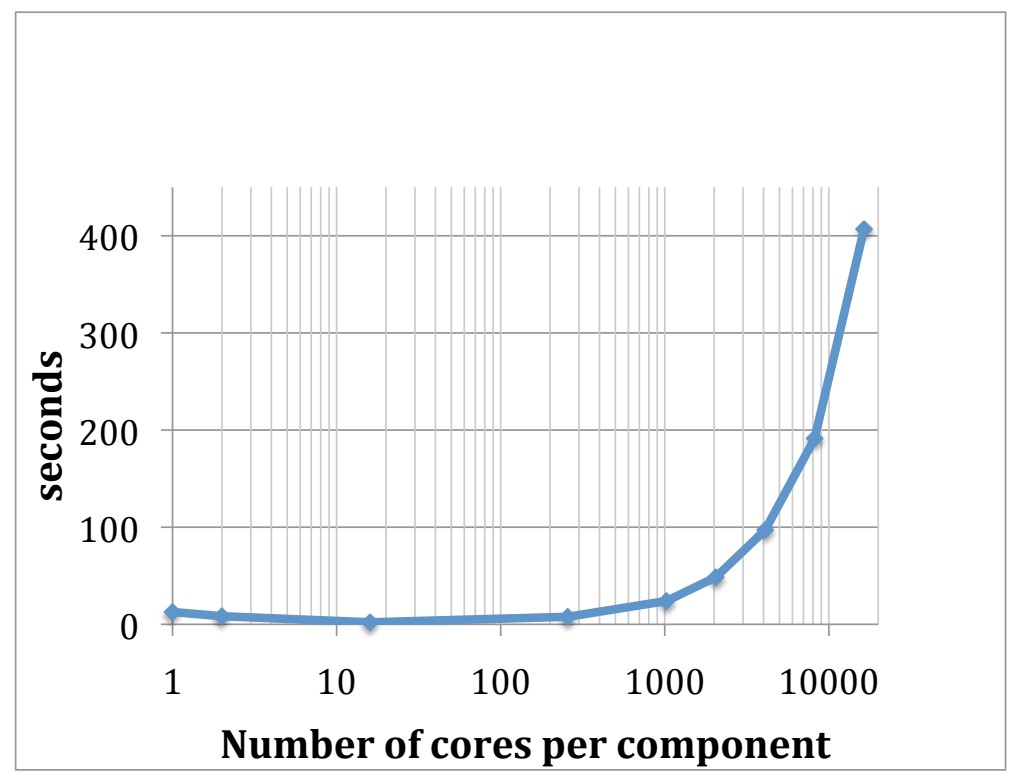

**Figure 3. Initialization cost for the T799-ORCA025 toy model using OASIS3-MCT_3.0 on Curie Bullx.**





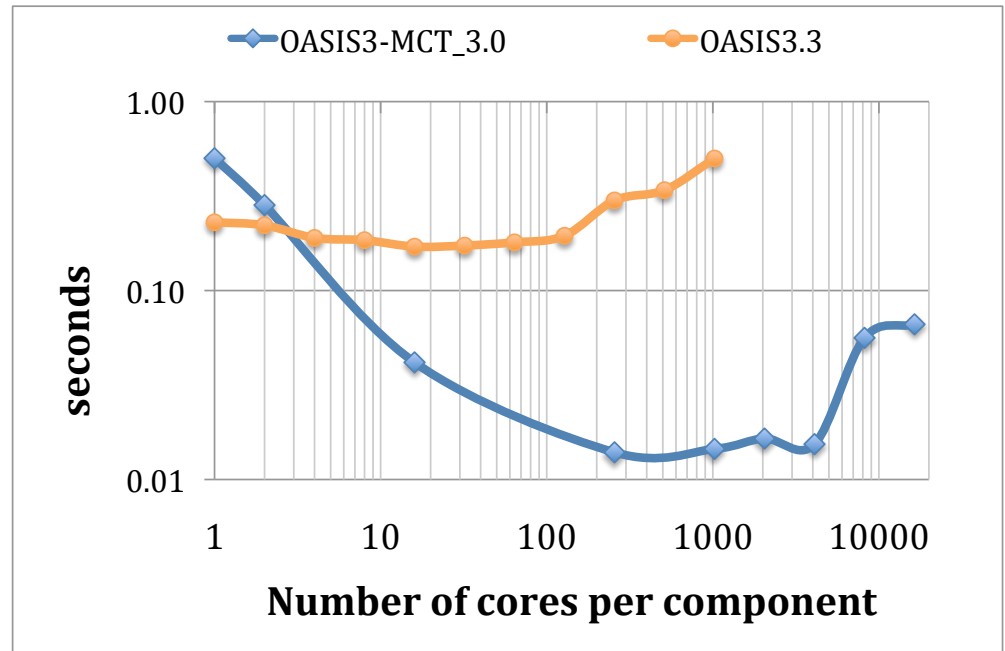

**Figure 4. Ping-pong time for the T799-ORCA025 toy model for OASIS3.3 and OASIS3-MCT_3.0 on Curie Bullx.**





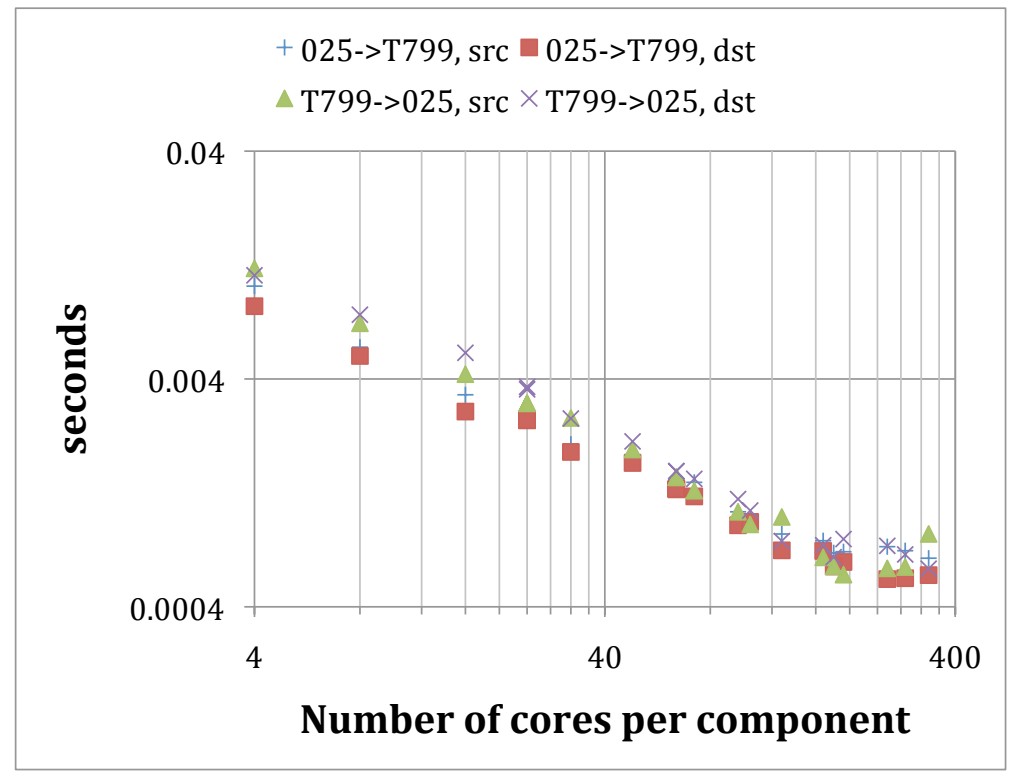

**Figure 5.  OASIS3-MCT_3.0 T799-ORCA025 mapping time versus core count per component on Lenovo.  *src*
and *dst* mapping are shown for both mapping directions using the *bfb* algorithm.**

| a) ORCA 025 cores | b) T799 cores | c) pipo time for mapping on *src* cores (s) | d) pipo time for mapping on *dst* cores (s) | e) pipo time for mapping on ORCA025 cores (s) | f) pipo time for mapping on T799 cores (s) |
|---|---|---|---|---|---|
| 24 | 336 | 5.10 | 5.48 | 7.29 | 3.79 |
| 180 | 180 | 1.29 | 1.54 | 1.36 | 1.36 |
| 336 | 24 | 4.70 | 4.93 | 1.91 | 6.69 |

**Table 1.  Comparison of the ping-pong (pipo) time for the T799-ORCA025 toy model on Lenovo on 360 cores
with both the relative core-count/component and the mapping location varied.  The time is in seconds for 1000
ping-pongs.  Columns a and b define the core-count used for each component of the toy model.  Columns c-f are**



the pipo times for 4 different mapping approaches c) mapping always on the source cores, d) mapping always on the destination cores, e) mapping on the ORCA025 cores, and f) mapping on the T799 cores.

| mapping time mapping = *src+bfb* | 1 field, 1 coupling | 10 fields, 10 couplings | 10 fields, 1 coupling | 10 fields, 1 bundle |
|---|---|---|---|---|
| ORCA025->T799 | 0.69 | 5.95 | 4.70 | 4.86 |
| T799->ORCA025 | 0.60 | 6.02 | 4.60 | 4.97 |

**Table 2. Comparison of mapping time for the T799-ORCA025 toy model on Lenovo on 180 cores per component for coupling of 1 field, coupling of 10 fields one at a time, coupling 10 fields using OASIS3-MCT multiple-coupling-field capability, and coupling of 10 fields by a single 3-D bundle. All times are for *src+bfb*
mapping on 1000 ping-pongs.**

| pipo time | 1 field, 1 coupling | 10 fields, 10 couplings | 10 fields, 1 coupling | 10 fields, 1 bundle |
|---|---|---|---|---|
| mapping = *src+bfb* | 1.29 | 10.56 | 11.89 | 12.15 |
| mapping = *dst+bfb* | 1.55 | 12.22 | 11.18 | 11.57 |

**Table 3. Comparison of ping-pong times for the T799-ORCA025 toy model setup on Lenovo on 180 cores per component for coupling 1 field, coupling 10 fields one at a time, coupling 10 fields via a single multiple-field coupling, and coupling of 10 fields by a single 3-D bundle for both *src* and *dst* mapping using the *bfb* mapping option. Times are accumulated over 1000 ping-pongs.**

| pes, mapping | CONSERV off | CONSERV *opt* | CONSERV *bfb* |
|---|---|---|---|
| 48, *src+bfb* | 4.04 | 7.56 | 177.24 |
| 48, *dst+bfb* | 4.41 | 7.33 | 176.75 |
| 180, *src+bfb* | 1.29 | 2.11 | 178.49 |
| 180, *dst+bfb* | 1.61 | 2.17 | 177.83 |

**Table 4. Comparison of ping-pong times for the T799-ORCA025 toy model on Lenovo on 48 and 180 tasks per model with the CONSERV option off, set to *bfb*, and set to *opt*.**




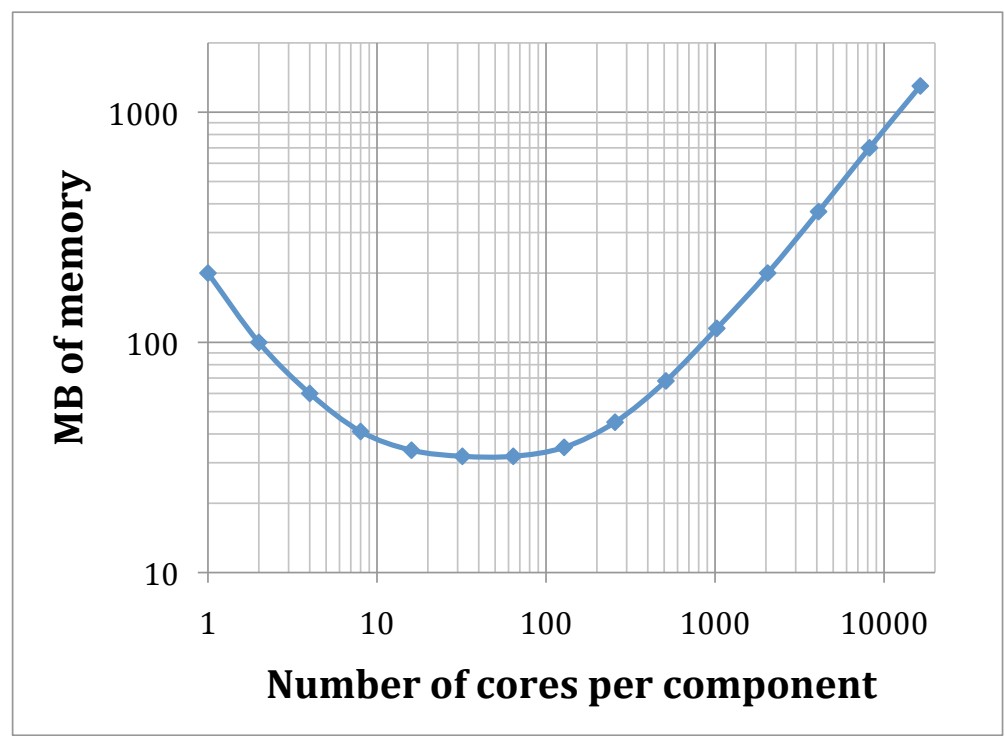

**Figure 6. OASIS3-MCT_3.0 memory use on Curie Bullx for the T799-ORCA025 toy model as a function of core-count/component.**