# Peer review of "Development and performance of a new version of the OASIS coupler, OASIS3-MCT\_3.0"

_Geoscientific Model Development, 2017_

## Referee Comment (RC1) · M. Hanke (Referee) · 21 Apr 2017

**General Comments**

This paper introduces the new version of the coupling software OASIS and its latest revision OASIS3-MCT_3.0. It describes in detail the most important improvements and new features of this version. In addition, it provides performance data relevant for users of the software.

It has a clear structure and is written well. It gives users of older versions of the software a good understanding of the changes and helps to decide whether to switch to the latest version or not. For developers of other coupling solutions this paper gives an interesting insight on how the current version of OASIS works.

After some modifications and clarifications regarding the presented performance results, I would recommend this paper for publication.

**Specific Comments**

If you are not familiar with coupling software in general or with OASIS is particular, some parts of the paper may be difficult to understand, due to usage of domain-specific terms and concepts without further explanation for example "hub coupler" in abstract, "top-level driver" in introduction, the terms "source" and "destination", "MCT router" in 2.1 General Architecture, or "CONSERV transform" in 2.4 Conservation. Depending on the target audience this might be an issue.

In the paper you talk about OASIS3-MCT and its improvements compared to older OASIS versions and about its latest revision OASIS3-MCT_3.0 in particular. However this is not reflected in the title of the paper. It implies that the paper is mainly about OASIS3-MCT_3.0.

You use lower and upper case when referencing figures or tables. This should be consistent.

"2.5 Concurrency, Process Layout, and Sequencing"
I do not see why there is a need to differentiate between different executables. Since each MPI process only has a single component, shouldn't it be enough to start the differentiation at the component level? This might reduce the complexity of this paragraph. Or would there be any difference if comp2, comp3, and comp4 were run on three individual executables?

The main conclusion of section "3.3 Interpolation" is that the default option of performing the mapping on the processes of the source component might not always be the best choice and that explicitly setting OASIS to do it on the processes of the component with the most resources can deliver better results. However, to draw this conclusion the presented test cases and diagrams seem to be overly complicated. Since the mapping is done based on a "simple one-dimensional" decomposition, the performance should be independent of the grid types being used. Therefore you could draw the same conclusion from a table similar to the following one (only showing the results for a one directions data exchange), which I think is much easier to understand:

| # src cores | # dst cores | Mapping on src | | Mapping on dst | |
|---|---|---|---|---|---|
| | | transfer | mapping | transfer | Mapping |
| 24 | 336 | * s | * s | * s | * s |
| 180 | 180 | * s | * s | * s | * s |
| 336 | 24 | * s | * s | * s | * s |

In the discussion of section "3.3 Interpolation", I would add that depending on where the mapping is executed, the amount of data that is exchanged between both components varies. This might be important in case both grids have a significantly different number of cells.

In the text it is nowhere mention what the abbreviation OASIS stands for.

P1L13-15 "It includes […] full parallelisation of the […] grid interpolation"
This may be interpolated as OASIS being able to generate interpolation weights on-the-fly in parallel.

P2L21 "source neighbour weights"
I do not know this term.

P3L7-8 "the hub coupler […] is no longer required"
This could be interpreted as: not required but still usable. Is that intended?

P4L15-18 "Compared to OASIS3 which required two data rearranges to couple fields in order to pass through the hub, OASIS3-MCT requires just one parallel rearrange to move data between two components."
You are comparing the coupling of fields in OASIS3 with the moving of data between components in OASIS3-MCT, which seems unfair, because in the paragraph above it is said that OASIS3-MCT also requires two data rearranges for the full coupling. Or is there a misunderstanding?

P5L9 "Mapping weight files can either be read directly"
For big weight files it may be important to know whether this is done in serial or in parallel. Only in section "4. Conclusions" it is mentioned that I/O in general is done in serial.

P5L18 "Users also have an additional option to specify the type of mapping to be carried out."
The term "type of mapping" is a little bit ambiguous. It could also refer to interpolation types (e.g. linear, nearest neighbour, or conservative interpolation).

P6L1-11 Maybe you should mention that there is the possibility to turn off the CONSERV transform. Which is important since this operation does not make sense for all field types.

P6L1-11 There is a bfb option for CONSERV transform and for mapping type. This can be confusing. Maybe clarify this.

P7L8-20 whole paragraph + Figure 2
This paragraph and the associated figure seem to be out of place. I would expect them to be part of a user manual.

P7L27 "a field put routine must be called before the matching get"
In case there are two components comp1 and comp2, if there is only a one directional data flow from comp1 to comp2, do all puts in comp1 actually have to be called before (in time) the respective gets in order to avoid a deadlock? Or do the gets wait until the respective put is called?

P8L11 "16,000 cores"
Maybe you should talk about MPI processes or specify that you are using one MPI process per core.

P8L28-29 "There is however clearly some concern that as core counts continue to increase, the initialization time will continue to grow."
Did you analyse the cause for the increase? Can you add some discussion on this?

P9L8-10 With two-nearest-neighbour interpolation you should have two weights per point on the destination grid.
T799->ORCA025: 2 * 1442 * 1021 = 2,944,564 weights << 4.5 mio weights
ORCA025->T799: 2 * 843,490 = 1,686,980 weights << 3 mio weights
Did I misunderstand something or how do you explain the difference in the number of weights?

P9L13-14 "above 8000 cores per component, the timing is degraded relative to lower core counts. At higher core counts, the timing depends heavily on the MPI performance."
Why do you not see this behaviour in the IS-ENES2 coupling technology benchmark? Is this due to the different grids used in both test cases?

P10L25 "while for the src+bfb case, the single operation performs slightly worse"
Are you sure that the measurements (10.56 vs 11.89), this statement is referring to, are correct?
(5.95 + 6.02) > 10.56 (taken from Table 2 and Table 3)

P11L13-14 "It's likely that the MPI memory footprint is accounting for most of this behavior (Balaji et al, 2008, Gropp, 2009)."
With a modern MPI implementation this should not happen. I have not seen this behaviour in similar measurements for the ICON model. You could verify this using for example the valgrind tool Massif.

P20 Figure 4
In this case, I would not use a trendline or and any line between the measurement points. The number of cores has a significant impact on the decomposition, which might lead to interesting result between the provided measurements. Therefore, a line between the points implies a continuity that might not reflect reality for this test case.

P21 Figure 5
Are these single measurements or averages?

P22 Table 2 and 3
(0.69 + 0.60) == 1.29 => Did the data exchange between the components only take a negligible amount of time?

P22 Table 2 and 3
(5.95 + 6.02) > 10.56 => Are the measurements correct?

P22 Table 2 and 3
(11.89 − (4.70 + 4.60)) > (12.15 − (4.86 + 4.97)) => time for mapping ↑ time for transfer ↓ => How do you explain this?

P22 Table 4
(2.11 − 1.29) >> (2.17 − 1.61)
I would assume that the cost for CONSERV is independent of the src/dst option. How do you explain the difference?

**Technical Comments**

P1L14 "separate hub coupler **process**"

P1L23 "OASIS is **a** coupling software"

P1L32-33 "OASIS-MCT supports coupling of fields on relatively arbitrary grids […]."
Is "OASIS-MCT supports coupling of fields defined on most grid types, commonly used in climate science, […]." better?

P1L33 "via a put/get approach. This approach means components make subroutine calls […]"
"via a put/get approach, which is based on components  making subroutine calls […]"?

P2L20-21 "calculation of the  weights and addresses needed for the mapping"

P2L23-26 check use of Oxford comma

P2L25 Why did you use the long form for AWI while using the abbreviation for ECMWF, KNMI, and MPI-M?

P2L26-28 Maybe add a reference?

P2L29-30 "OASIS3-MCT extended the widely used and distributed OASIS3 version of the model."
"It extends the widely used and distributed OASIS3."?

P3L8 "Transform**ation**s are carried out"

P3L21 "section 4 provides a summary"
Section 4 is called "Conclusion"

P6L2-3 "In OASIS3-MCT, this operation **can**  now **be** performed in parallel on the source or destination processes"
If the bfb option is used, it will still be done in serial, or not?

P7L1-2 "are indicated by the different lettered arrows ."

P9L10 ""

P10L5 "(1.91**s** vs 4.70**s**)"

P10L32 "CONSERV unset"
In Table 4 this is called off.

P10L31-33 "Table 4 shows [...]. Table 4 shows [...]"
Identical start of two consecutive sentences.

P11L4-5 "such as area overlap conservative"
Maybe place a reference to:
 http://dx.doi.org/10.1175/1520-0493(1999)127%3C2204:FASOCR%3E2.0.CO;2

P12L19 "tens of thousands"

P12L27-28 "best performance"

P16L5-7 "Valcke […] 2012a"
P16L12-14 "Valcke […] 2015"
Could not find references of these papers in the text.

P19 Figure 3
P20 Figure 4
P21 Figure 5
x-axis: maybe use logarithmic base 2 instead of 10
y-axis label: "**Time in s**"

P19 Figure 3
y-axis: use logarithmic scale to better show behaviour for 1 to 1000 cores per component

P21 Figure 5
The data set "T799->025,dst" seems to have two data points at 24 core per component while all others only have one.

P21 Figure 5
For higher number of cores (> 40), the choice of the symbols for the individual data sets makes it hard to read.

P22 Table 4 "**cores**"

P22L23 "**cores**"

P23 Figure 6
x-axis: maybe use logarithmic base 2 instead of 10

P23 Figure 6
MB or MiB? per core?

**Questions not necessarily relevant for the paper**
P1L19-20 "10,000 two dimensional coupling fields"
In case of 3d fields, would the different levels be counted as separate fields?

P7L24 "OASIS3-MCT provides some new capabilities to detect potential deadlocks before they occur"
Very interesting! Can you be more specific?

P7L28-29 "In OASIS3-MCT, puts are always non-blocking while gets are blocking."
Are there plans for non-blocking gets?

P12L11-12 "the cost associated with generating the mapping files can be moved to a preprocessing step"
Which not necessarily has to be faster, if weight computation is done in parallel.

---

## Referee Comment (RC2) · Anonymous Referee #2 · 27 Apr 2017

General Comments:

This paper provides a good overview of the latest version of OASIS, OASIS3-MCT_3.0, which has a series of new capabilities, such as the usage of offline mapping weights, the ability of coupling intra one executable, and parallel data coupling without separate processes. OASIS3-MCT has already been tested with different configurations on up to 32,000 processes, and has already been used in the community. I recommend to publish this paper after addressing the following specific comments.

Specific comments:

1. The abstract (as well as some other parts in the context, such as P3 L14∼L15, P6 L14∼L15, and P11L31) mentions that "OASIS3-MCT_3.0 is the latest release and includes the ability to couple between components running sequentially on the same

set of tasks". It seems contradictory to P6 L24∼L25 that "Each task will be associated with only one executable and one component in any application", which indicates that components cannot share any task. According to the API of "oasis_init_comp", I think the statement in P6 L24∼L25 is true.

2. P1 L15∼L18, P6 L18∼L19, P6 L25∼L27 and P12 L1∼L2 may indicate that that there can be two different decompositions of the same grid within the same component and these two decompositions can have different subsets of the tasks (processes). To achieve this capability, the API "oasis_def_partition" has been extended with an additional parameter "name". When I read the user manual at the first time, I guested that "name" means the name of the grid. After a careful consideration, I think that "name" should be the keyword of a decomposition but not the name of the corresponding grid, which means that the "name" corresponding to two different decompositions of the same grid within the same component should be different. If that point is true, please clarify it.

3. The ability to define grids has been mentioned several times in the paper. What does it mean when only the API for writing grid data into files are introduced in the user manual. According to Figure 2, is the grid defined implicitly in the definition of decomposition?

4. Compared to OASIS3, OASIS3-MCT_3.0 have a new capability of pre-defined mapping files. After reading the paper as well as the user manual, it is still unclear for me that how to make OASIS3-MCT_3.0 know which mapping file should be used for a specific set of coupling fields (for example, users may want to use bilinear algorithm for state fields and use conservative algorithm for flux fields when coupling fields from an atmosphere model to an ocean model). Is there any restriction when users using the pre-defined mapping file. Concrete examples are welcome for this new capability.

5. P7 L28∼L29. It is interesting to know how to make the puts non-blocking. In MCT, the data sending is blocking for example with the MPI_wait, which indicates that such
MPI_wait should be disabled for the non-blocking puts. It seems that OASIS3-MCT_3.0 does not use another MPI_wait out of MCT. So, one interesting question here is that how OASIS3-MCT_3.0 guarantees the puts constantly non-blocking (for example, we encountered the case that MPI_Isend was blocked when we sent a large message or many small massages) and how OASIS3-MCT_3.0 achieves safe non-blocking puts (for example, how to guarantee that next puts do not flush the data of previous puts in memory buffer).

6. P6 L10~L11 states that "The opt option will however be bit-for-bit reproducible if the same number of processes is used between different runs". Given the same number of processes, bit-for-bit results may fail to be reproduced if the decomposition changed.

7. One suggestion regarding Section 2.4 is that the opt option can use higher-precision of floating-point calculation to achieve faster bit-for-bit identical reduction. For example, using REAL8 when the coupling fields are REAL4 and using REAL16 when coupling fields are REAL8.

8. Some results in Table 4 seem strange to me. Why the time for <10 fields, 10 couplings> is obviously smaller than 10 times of the time of <1 field, 1 coupling>? Why <10 fields, 1 coupling> is not much faster than <10 fields, 10 couplings>? The most significant reason may be the MPI message size of <1 field, 1 coupling> is big because the two components have similar decompositions and the core number is small relative to the big grid size. Given the same core number, more test cases with smaller grid size and different decompositions between the two components are welcome.

9. The year of the first reference should be 2008.
* * *

---

## Referee Comment (RC3) · Anonymous Referee #3 · 10 May 2017

This paper describes a new version of the OASIS coupler, called OASIS3-MCT, following upon other papers documenting earlier versions.

The key advance here is the inclusion of the MCT (Model Coupling Toolkit) as a layer providing parallel regridding services. The paper convincingly demonstrates the benefits of the MCT layer in computing interpolation weights online and offline. The analysis is quite comprehensive, and covers the technical aspects of coupling and interpolation. It explores the performance aspects of choosing to perform the computationally expensive operations on the source or destination side, or on dedicated core(s). It covers the issues of various coupling timestepping algorithms and parallel layouts, with different arrangements of components in serial and parallel. The tests are well-designed and representative of very high-end models. They have been performed with some toy models, but with resolutions set to the highest-end models of today, and at challenging

core counts.

Links have been provided to ensure public availability of code.

The paper is very well-written with no obvious typos or poor sentence construction.

I recommend publication as-is, as it is a complete description of an existing piece of software. The comments below are what I found to be omissions in the paper, but perhaps should be treated as suggestions for future versions.

1) There is no discussion of OpenMP as an alternative to MPI. Future hardware will require going to more shared memory and less message passing.

2) There is no discussion of GPUs, MICs, etc and plans to port OASIS to novel architectures.

3) I am somewhat taken aback by the extreme cost of providing bfb (bit for bit) reproducing algorithms. In other similar codes this cost ratio is somewhat lower (which could of course mean that the non-reproducing modes in other codes are too slow!) This may require some work.

———————————————

---

## Referee Comment (RC4) · M. Hanke (Referee) · 11 May 2017

In RC3 Referee #3 wrote: "The paper convincingly demonstrates the benefits of the MCT layer in computing interpolation weights online and offline."

As far as I understood the paper, MCT is not at all involved in the online weight computation and it is not explicitly mentioned for the offline weight computation.

---

## Author Comment (AC2) · 2 Jul 2017

Reply to Anonymous Referee #2.

We would like to thank referee #2 for taking the time to review our paper and for the thoughtful comments. We will reply to each comment individually below.

Specific comments:

1. The abstract (as well as some other parts in the context, such as P3 L14~L15, P6 L14~L15, and P11L31) mentions that "OASIS3-MCT_3.0 is the latest release and includes the ability to couple between components running sequentially on the same set of tasks". It seems contradictory to P6 L24~L25 that "Each task will be associated with only one executable and one component in any application", which indicates that components cannot share any task. According to the API of "oasis_init_comp", I think the statement in P6 L24~L25 is true.

We have clarified the sentence in the introduction to "OASIS3-MCT_3.0 extends the ability to couple components running concurrently and adds support for coupling within a component for grids and fields defined on overlapping or partially overlapping sets of tasks, such as between physics and dynamics modules within an atmospheric model or to and from a model I/O module." We have clarified the description a bit in section 2.5. In particular, we have updated the first sentence to be " The ability to couple fields within one executable running on partially overlapping tasks was added in OASIS3-MCT_3.0". We have also added a sentence, "While OASIS3-MCT supports both single and multiple executable configurations, the coarsest level of concurrency in the system is the component." In the conclusions, we modified the sentence to " OASIS3-MCT_3.0 also provides new capabilities to couple fields within a single component running on concurrent, overlapping, or partially overlapping processes ". The reviewer makes a good point that we were implying that components could run on overlapping tasks and that's not true and that has been fixed in the text.

2. P1 L15~L18, P6 L18~L19, P6 L25~L27 and P12 L1~L2 may indicate that that there can be two different decompositions of the same grid within the same component and these two decompositions can have different subsets of the tasks (processes). To achieve this capability, the API "oasis_def_partition" has been extended with an addi- tional parameter "name". When I read the user manual at the first time, I guested that "name" means the name of the grid. After a careful consideration, I think that "name" should be the keyword of a decomposition but not the name of the corresponding grid, which means that the "name" corresponding to two different decompositions of the same grid within the same component should be different. If that point is true, please clarify it.

That is correct, the name associated with the "oasis_def_partition" call is the name given to the parition, not to the grid. We will clarify in the user guide.

3. The ability to define grids has been mentioned several times in the paper. What does it mean when only the API for writing grid data into files are introduced in the user manual. According to Figure 2, is the grid defined implicitly in the definition of decomposition?

The grid is something that does not depend on the decomposition and defines the grid center, corner, area, and mask information. At run time, OASIS reads this grid in a file that can be either produced by the user before the run or written through the API by the model. A partition is specific decomposition of a grid in the model. We have removed figure 2 from the revised draft as this better fits into the user guide and we will update the user guide to clarify.

4. Compared to OASIS3, OASIS3-MCT_3.0 have a new capability of pre-defined mapping files. After reading the paper as well as the user manual, it is still unclear for me that how to make OASIS3-MCT_3.0 know which mapping file should be used for a specific set of coupling fields (for example, users may want to use bilinear algorithm for state fields and use conservative algorithm for flux fields when coupling fields from an atmosphere model to an ocean model). Is there any restriction when users using the pre-defined mapping file. Concrete examples are welcome for this new capability.

We will clarify this information in the user guide. For a given entry in the namcouple file, the namcouple keyword MAPPING specifies the mapping file for those coupling fields. Each coupling field can be associated with a different mapping file rather arbitrarily and each mapping file can be generated via different algorithms.

5. P7 L28~L29. It is interesting to know how to make the puts non-blocking. In MCT, the data sending is blocking for example with the MPI_wait, which indicates that such
MPI_wait should be disabled for the non-blocking puts. It seems that OASIS3-MCT_3.0 does not use another MPI_wait out of MCT. So, one interesting question here is that how OASIS3-MCT_3.0 guarantees the puts constantly non-blocking (for example, we encountered the case that MPI_Isend was blocked when we sent a large message or many small massages) and how OASIS3-MCT_3.0 achieves safe non-blocking puts (for example, how to guarantee that next puts do not flush the data of previous puts in memory buffer).

MCT supports non-blocking MPI. The reviewer is correct that at some point, MCT will execute an MPI_Wait for a non-blocking MPI_ISend. On the put side, this happens before the next put of the same data at the next timestep. We define this as non-blocking MPI because the model does not wait for the actual put to occur and

the model can continue to advance. In fact, the put is only non-blocking in the sense that it can be only one coupling period ahead of the get at the most. While on the get side, the MPI is blocking at the time of the get. We have clarified the text in section 2.5 to reflect this information.

6. P6 L10~L11 states that "The opt option will however be bit-for-bit reproducible if the same number of processes is used between different runs". Given the same number of processes, bit-for-bit results may fail to be reproduced if the decomposition changed.

The reviewer is correct that if the decomposition changes, the sum will not be bit-for-bit reproducible. We have updated that sentence as follows, " The *opt* option will however be bit-for-bit reproducible if the same number of processes and decomposition are used between different runs ." We have also updated the conclusions.

7. One suggestion regarding Section 2.4 is that the opt option can use higher-precision of floating-point calculation to achieve faster bit-for-bit identical reduction. For example, using REAL8 when the coupling fields are REAL4 and using REAL16 when coupling fields are REAL8.

We have significantly revised section 2.4 to include some preliminary results of three new global sum algorithms including the algorithm suggested by the reviewer that are currently in the development version of OASIS3-MCT and expected in the OASIS3-MCT_4.0 release. The global sum calculation implemented in OASIS3-MCT_3.0 needed significant revision as indicated in the earlier version of the paper and this has already been undertaken.

8. Some results in Table 4 seem strange to me. Why the time for <10 fields, 10 couplings> is obviously smaller than 10 times of the time of <1 field, 1 coupling>? Why <10 fields, 1 coupling> is not much faster than <10 fields, 10 couplings>? The most significant reason may be the MPI message size of <1 field, 1 coupling> is big because the two components have similar decompositions and the core number is small relative to the big grid size. Given the same core number, more test cases with smaller grid size and different decompositions between the two components are welcome.

We have merged and updated the results in table 3 and 4 and added some new information. We have added a barriered ping pong time to compare with an unbarried time. This provides additional insights into the results that were not available in the initial version of the paper. In particular, 10 fields, 10 couplings is fastest in the unbarriered ping-pong time because it seems the amount of work that is overlapped between coupling and mapping is highest in that case. That case has the highest performance degradation when the send and mapping are barriered and

the mapping time of the 10 fields, 1 coupling is faster.  These issues are now discussed in the paper in section 3.4.

9. The year of the first reference should be 2008.

We have changed 2009 to 2008, thanks.

---

## Author Comment (AC4) · 2 Jul 2017

[revised manuscript text omitted]

Tony Craig 6/15/17 11:40 AM

[Figure]

MPI tasks

0 1 2 3 4 5 6 7 8 9 10 11 12 13 14 15 16 17 18 19 20 21 22 23 24 25 26 27 28 29 30 31 32 33 34 35 36 37

exe1 - comp1 -grid1

exe2 − comp2 − grid2

exe2 − comp3

exe2 − comp3 − grid3

exe2 − comp3 − grid4

exe2 − comp4

exe2 − comp3 − grid5

A
B
C
D
E
F
G
H
I
J

5      **Figure 1. A schematic of the coupling capability in OASIS3-MCT_ 3.0.  In this example, there are 2 executables, exe1 and exe2.  Executable 2 has 3 components, comp2, comp3, and comp4, and comp3 has 3 grids, grid3, grid4, and grid5;  comp4 is not involved in any coupling in this case.  The executables, components, and grids are laid out across different tasks.  Arrows indicate different coupling capabilities; A), D), E), and J) between different components in different executables;  B), F), and I) in a single executable between different components with**
10     **different grids; C) between different grids in a single component on non-overlapping tasks; G) between different grids in a single component on partially overlapping tasks; and H) between different grids in a single component on partially overlapping and partially non overlapping tasks.**

Tony Craig 6/7/17 5:50 PM

Tony Craig 6/7/17 5:50 PM

oasis_put/get          oasis_put/get
...                    ...
oasis_put/get          oasis_put/get
oasis_terminate        oasis_terminate

[Figure]

**Figure 2. Initialization cost for the T799-ORCA025 toy model using OASIS3-MCT_3.0 on Curie Bullx.**

Tony 6/26/17 1:54 PM

[Figure]

**Figure 3. Comparison of the ping-pong (pipo) time for the T799-ORCA025 toy model for OASIS3.3 and OASIS3-MCT_3.0 on Curie Bullx. The time is averaged for a run where 1000 ping-pongs were carried out.**

Tony 6/26/17 1:54 PM
Tony 6/26/17 8:55 AM

[Figure]

**Figure 4. OASIS3-MCT_3.0 T799-ORCA025 mapping time versus core count per component on Lenovo.** *src* **and** *dst* **mapping are shown for both mapping directions using the** *bfb* **algorithm based on tests where 1000 ping-pongs were run.**

Tony 6/26/17 1:54 PM
Tony 7/1/17 2:33 AM

| a) ORCA 025 cores | b) T799 cores | c) pipo time for mapping on *src* cores (s) | d) pipo time for mapping on *dst* cores (s) | e) pipo time for mapping on ORCA025 cores (s) | f) pipo time for mapping on T799 cores (s) |
|---|---|---|---|---|---|
| 24 | 336 | 5.10 | 5.48 | 7.29 | 3.79 |
| 180 | 180 | 1.29 | 1.54 | 1.36 | 1.36 |
| 336 | 24 | 4.70 | 4.93 | 1.91 | 6.69 |

**Table 1. Comparison of the ping-pong (pipo) time for the T799-ORCA025 toy model on Lenovo on 360 cores with both the relative core-count/component and the mapping location varied. The time is in seconds for 1000 ping-pongs. Columns a and b define the core-count used for each component of the toy model. Columns c-f are**

the pipo times for 4 different mapping approaches c) mapping always on the source cores, d) mapping always on the destination cores, e) mapping on the ORCA025 cores, and f) mapping on the T799 cores.

| time (seconds) mapping = src+bfb | 1 field, 1 coupling | 10 fields, 10 couplings | 10 fields, 1 coupling | 10 fields, 1 bundle |
|---|---|---|---|---|
| pipo time, no barriers | 1.29 | 10.52 | 11.93 | 12.29 |
| pipo time, with barriers | 1.87 | 17.63 | 16.56 | 17.48 |
| map ORCA025->T799 with barriers | 0.67 | 5.48 | 4.61 | 4.68 |
| map T799->ORCA025 with barriers | 0.56 | 5.28 | 4.76 | 4.81 |

**Table 2. Comparison of unbarriered and barriered ping-pong (pipo) and barriered mapping time for the T799-ORCA025 toy model on Lenovo on 180 cores per component for coupling of 1 field, coupling of 10 fields one at a time, coupling 10 fields using OASIS3-MCT multiple-coupling-field capability, and coupling of 10 fields by a single 3-D bundle. All times are for src+bfb mapping for 1000 ping-pongs. For barriered times, MPI barriers were introduced in both components before the send and before the mapping to force serialization of work and to time the mappings separately.**

| cores mapping | CONSERV unset | CONSERV lsum8 | CONSERV lsum16 | CONSERV ddpdd | CONSERV reprosum | CONSERV gather |
|---|---|---|---|---|---|---|
| 48, src+bfb | 4.00 | 8.27 | 16.78 | 10.65 | 17.34 | 117.72 |
| 48, dst+bfb | 4.39 | 8.02 | 16.59 | 10.42 | 16.98 | 142.12 |
| 180, src+bfb | 1.25 | 2.21 | 4.59 | 2.87 | 4.85 | 126.91 |
| 180, dst+bfb | 1.56 | 2.26 | 4.62 | 2.92 | 4.90 | 130.01 |

**Table 3. Comparison of ping-pong (pipo) times for the T799-ORCA025 toy model on Lenovo on 48 and 180 cores per model with the CONSERV option off (unset), set to lsum8 (opt in OASIS3-MCT 3.0), lsum16, ddpdd, reprosum and gather (bfb in OASIS3-MCT). Times are accumulated over 1000 ping-pongs for a single coupling field in each direction.**

Tony 6/21/17 2:06 AM
Tony 6/21/17 2:07 AM
Tony 6/26/17 8:39 AM
Tony 6/26/17 8:53 AM
Tony 6/26/17 8:53 AM
Tony 6/26/17 8:53 AM
Tony Craig 6/7/17 5:36 PM
Tony Craig 6/7/17 5:08 PM
Tony Craig 6/16/17 3:40 PM
Formatted ... [36]
Tony Craig 6/16/17 3:37 PM
Tony Craig 6/16/17 3:40 PM
Formatted ... [37]
Tony 6/26/17 8:42 AM
Tony Craig 6/16/17 3:44 PM
Tony 6/26/17 8:43 AM
Tony Craig 6/16/17 3:44 PM
Tony 6/26/17 8:43 AM
Tony 6/26/17 8:42 AM
Tony Craig 6/16/17 3:44 PM
Tony 6/26/17 8:43 AM
Tony Craig 6/16/17 3:43 PM
Tony 6/26/17 8:43 AM
Tony 6/26/17 8:42 AM
Tony Craig 6/16/17 3:44 PM
Tony 6/26/17 8:43 AM
Tony Craig 6/16/17 3:43 PM
Tony 6/26/17 8:43 AM
Tony 6/26/17 8:42 AM
Tony Craig 6/16/17 3:44 PM
Tony 6/26/17 8:43 AM
Tony Craig 6/16/17 3:43 PM
Tony 6/26/17 8:43 AM
... [42]
Tony 6/26/17 1:52 PM
Tony Craig 6/7/17 5:37 PM
... [43]
Tony Craig 6/16/17 3:39 PM

[Figure]

**Figure 5. OASIS3-MCT_3.0 memory use on Curie Bullx for the T799-ORCA025 toy model as a function of cores per component.**

Tony 6/26/17 1:54 PM

Tony 7/1/17 2:28 AM

Tony 7/1/17 2:28 AM

Tony 7/1/17 2:28 AM

---

## Author Comment (AC5) · 2 Jul 2017

An updated manuscript with changes highlighted is available under RC1.

---

## Author Comment (AC6) · 2 Jul 2017

An updated manuscript with documented changes is available under the reply to RC1.

---

## Author Comment (AC1)

Reply to Referee #1.

We would like to thank Moritz Hanke for his careful review and thoughtful comments. We will reply to the comments below (in green text)

**General Comments**

This paper introduces the new version of the coupling software OASIS and its latest revision OASIS3-MCT_3.0. It describes in detail the most important improvements and new features of this version. In addition, it provides performance data relevant for users of the software.

It has a clear structure and is written well. It gives users of older versions of the software a good understanding of the changes and helps to decide whether to switch to the latest version or not. For developers of other coupling solutions this paper gives an interesting insight on how the current version of OASIS works.

After some modifications and clarifications regarding the presented performance results, I would recommend this paper for publication.

**Specific Comments**

If you are not familiar with coupling software in general or with OASIS is particular, some parts of the paper may be difficult to understand, due to usage of domain-specific terms and concepts without further explanation for example "hub coupler" in abstract, "top-level driver" in introduction, the terms "source" and "destination", "MCT router" in 2.1 General Architecture, or "CONSERV transform" in 2.4 Conservation. Depending on the target audience this might be an issue.

We have added the following clarifications:
- " A separate top-level driver to control system sequencing is not required "
- " all coupling fields passed through a separate central hub coupler component"
- source and destination are implicitly defined in the introduction
- "Each parallel field in the source model was gathered to a single process on the hub where operations such as mapping and time averaging were executed, and the field was then scattered to the destination model",
- MCT router is an MCT datatype. We have updated the text and the only place where "router" appears in text is in the following sentence where it is defined, " Data communication and mapping rearrangement is handled internally in OASIS3-MCT via MCT routers.".
- "CONSERV" is clearly defined in section 2.4, "The CONSERV operation computes global sums of the source and destination fields and applies corrections to the decomposed mapped field in order to conserve area-integrated field quantities."

In the paper you talk about OASIS3-MCT and its improvements compared to older OASIS versions and about its latest revision OASIS3-MCT_3.0 in particular. However this is not reflected in the title of the paper. It implies that the paper is mainly about OASIS3-MCT_3.0.

To be honest, the original title of the paper was " Development and performance of a new version of the OASIS coupler, OASIS3-MCT ", but the editor encouraged us to be more specific with regard to the version in the title prior to formal submission. The paper is written at a time when OASIS3-MCT_3.0 is the current release, and so we feel it is reasonable to include that information in the title. It is true that this paper takes a slightly broader approach by summarizing changes since OASIS3 including features added before OASIS3-MCT_3.0 (see details in Appendix A). It even includes some information about what is coming in the version 4.0 release of OASIS3-MCT. We made a few changes in the text to further clarify the scope of the paper but feel the current title is reasonable. In particular, we have added " This paper describes the development of OASIS3-MCT from OASIS3 to the current version 3.0 release and will also introduce some new features expected in the version 4.0 release." to the introduction.

You use lower and upper case when referencing figures or tables. This should be consistent.

We have updated the text so all references to figures and tables in the text are lower case unless they occur at the start of the sentence.

"2.5 Concurrency, Process Layout, and Sequencing"
I do not see why there is a need to differentiate between different executables. Since each MPI process only has a single component, shouldn't it be enough to start the differentiation at the component level? This might reduce the complexity of this paragraph. Or would there be any difference if comp2, comp3, and comp4 were run on three individual executables?

The reviewer's comments are correct. It doesn't fundamentally matter whether multiple components are run as a single executable or as multiple executables in Oasis3-MCT. I think the main point of including that statement is to make it clear that both modes are supported. We have added a sentence at the end of the second paragraph in section 2.5 to emphasize that point and address the reviewer's concerns.

The main conclusion of section "3.3 Interpolation" is that the default option of performing the mapping on the processes of the source component might not always be the best choice and that explicitly setting OASIS to do it on the processes of the component with the most resources can deliver better results. However, to draw this conclusion the presented test cases and diagrams seem to be overly complicated. Since the mapping is done based on a "simple one-dimensional"

decomposition, the performance should be independent of the grid types being used. Therefore you could draw the same conclusion from a table similar to the following one (only showing the results for a one directions data exchange), which I think is much easier to understand:

| # src cores | # dst cores | Mapping on src | | Mapping on dst | |
|---|---|---|---|---|---|
| | | transfer | mapping | transfer | Mapping |
| 24 | 336 | *s | *s | *s | *s |
| 180 | 180 | *s | *s | *s | *s |
| 336 | 24 | *s | *s | *s | *s |

In the discussion of section "3.3 Interpolation", I would add that depending on where the mapping is executed, the amount of data that is exchanged between both components varies. This might be important in case both grids have a significantly different number of cells.

This is a reasonable point. However, Figure 5 is useful in that it shows the scaling of mapping across a broader range of pe counts which some readers might find useful. The other problem is that while it's relatively easy to time the mapping separately with appropriate barriers, it's much harder to time the transfer in these cases as there is significant load imbalance, puts are non-blocking, and some of the performance is associated with overlapping transfer and mapping work. We believe the information in table 1 is consistent with the reviewers request, and figure 5 provides additional insight into the mapping performance that goes beyond what could be done with a table. A final point is that it's not correct to suggest the map timing is independent of grid type. In fact, the grid decomposition, number of weights, whether mapping is done on the src or dst side, number of pes in play, and distribution of the weights have a large impact on the map timing. We have updated figure 5, so the symbols and symbol key are clearer. We do agree about the comment that amount of data exchanged is important and we have added the statement, " Another point is that if there is a large disparity in the number of grid cells in the two mapped grids, it should be better to exchange the coupling fields expressed on the grid with the fewest grid cells and perform the remapping on the other component tasks."

In the text it is nowhere mention what the abbreviation OASIS stands for.

We have added a sentence in the first paragraph of the introduction to define the OASIS project.

P1L13-15 "It includes [...] full parallelisation of the [...] grid interpolation"
This may be interpolated as OASIS being able to generate interpolation weights on-the-fly in parallel.

We have changed the sentence to read, "parallelization of the coupling communication and run time grid interpolation " to emphasize parallelization of the interpolation at run time, which is unrelated to the process of weights .

P2L21 "source neighbour weights" I do not know this term.

We have rewritten this sentence as "In particular, OASIS4 included a library that performed a parallel calculation for generation of the mapping weights and addresses needed for the interpolation of the coupling fields."

P3L7-8 "the hub coupler [...] is no longer required"
This could be interpreted as: not required but still usable. Is that intended?

This is a good point. We have changed this sentence to "Third, the OASIS hub coupler was deprecated and is no longer needed or implemented."

P4L15-18 "Compared to OASIS3 which required two data rearranges to couple fields in order to pass through the hub, OASIS3-MCT requires just one parallel rearrange to move data between two components."
You are comparing the coupling of fields in OASIS3 with the moving of data between components in OASIS3-MCT, which seems unfair, because in the paragraph above it is said that OASIS3-MCT also requires two data rearranges for the full coupling. Or is there a misunderstanding?

This is a very good point. We have clarified this sentence as follows, " Compared to OASIS3, which required an all-to-one communication, interpolation on the single hub process, and a one-to-all communication to couple fields, OASIS3-MCT requires just one parallel all-to-all communication between the source and destination processes and one parallel mapping which includes a rearrangement of the data on the source or destination processes. " We have also changed some of the wording in the document to provide more consistency, clarifying the terms redistribution, communication, coupling, and mapping.

P5L9 "Mapping weight files can either be read directly"
For big weight files it may be important to know whether this is done in serial or in parallel. Only in section "4. Conclusions" it is mentioned that I/O in general is done in serial.

We have added a new sentence to further define the implementation, "In OASIS3-MCT, the weight files are read serially on the root process and distributed to other processes in reasonable chunks. That chunk size is currently set to 100,000 weights at a time to limit memory use on the root process."

P5L18 "Users also have an additional option to specify the type of mapping to be carried out." The term "type of mapping" is a little bit ambiguous. It could also refer to interpolation types (e.g. linear, nearest neighbour, or conservative interpolation).

This is a good comment. We have changed this sentence to "Users also have an additional option to set the implementation of the underlying mapping algorithm."

P6L1-11 Maybe you should mention that there is the possibility to turn off the CONSERV transform. Which is important since this operation does not make sense for all field types.

We have added the word "optional" in the first sentence of section 2.4 to reiterate the fact that CONSERV is an optional transform. We have also updated this section to reflect some new features.

P6L1-11 There is a bfb option for CONSERV transform and for mapping type. This can be confusing. Maybe clarify this

We recognize that the common keywords are not ideal and are working to differentiate them in future releases. We have added a sentence in section 2.4 to clarify, "Note that both the CONSERV operation and the underlying mapping algorithm setting share a common flag, *bfb*, but that these two settings are completely independent."

P7L8-20 whole paragraph + Figure 2
This paragraph and the associated figure seem to be out of place. I would expect them to be part of a user manual.

We have removed this section and Figure 2 from the paper. This information is in the user guide and we agree that this does not need to be duplicated in the paper.

P7L27 "a field put routine must be called before the matching get"
In case there are two components comp1 and comp2, if there is only a one directional data flow from comp1 to comp2, do all puts in comp1 actually have to be called before (in time) the respective gets in order to avoid a deadlock? Or do the gets wait until the respective put is called?

This is a good question and something we've been trying to clarify in the implementation and user guide. To answer the question, each put is non-blocking but waits for the completion of the put of the same coupling field at the previous coupling timestep before it executes. Therefore, you cannot queue up a bunch of puts before executing a get on overlapping or non-overlapping pes. We have tried to clarify this paragraph in section 2.5 by adding, "In OASIS3-MCT, puts are generally non-blocking while gets are blocking. More specifically, a put waits for the completion of the put of the same coupling field at the previous coupling timestep before proceeding in order to prevent puts from queuing up in MPI and using excess memory. In other words, for a specific put-get pair, the last put can never be more than one coupling period ahead of the equivalent get in OASIS3-MCT. This means that the puts and gets have to be interleaved when coupling on overlapping tasks. It

is not possible to queue up a series of puts over multiple coupling periods before executing the equivalent gets."

P8L11 "16,000 cores"
Maybe you should talk about MPI processes or specify that you are using one MPI process per core.

We are constantly struggling whether to use MPI tasks, processes, cores, or pes as a way to describe parallelism. We have tried to be consistent in the paper. We have changed the text from "16,000 cores" to "16,000 MPI tasks".

P8L28-29 "There is however clearly some concern that as core counts continue to increase, the initialization time will continue to grow."
Did you analyse the cause for the increase? Can you add some discussion on this?

To address this comment, the end of the last paragraph in section 3.1 has been updated as follows, "The initialization uses MPI heavily to initialize the coupling interactions, read in the mapping files, and setup the communication for the mapping rearrangement and coupling communication. In general, the initialization is not expected to scale well, but the initialization overhead is what allows the model to run efficiently during the actual run phase. There is clearly some concern that as core counts continue to increase, the initialization time will continue to grow. OASIS developers continue to monitor and analyze both the runtime and initialization costs and make improvements. "

P9L8-10 With two-nearest-neighbour interpolation you should have two weights per point on the destination grid.
T799->ORCA025: 2 * 1442 * 1021 = 2,944,564 weights << 4.5 mio weights
ORCA025->T799: 2 * 843,490 = 1,686,980 weights << 3 mio weights
Did I misunderstand something or how do you explain the difference in the number of weights?

We had an error in the description, the weights are based on five-nearest-neighbor interpolation and the ORCA025 grid has masked points. 4.5 million weights for T799->ORCA025 is the equivalent of 61% unmasked points on the ocean grid, 3.0 million weights for ORCA025->T799 is 71% of the maximum number of weights if the grids were unmasked. We have corrected that section and it now reads, "Each coupling of data between a pair of components consists of a mapping operation that interpolates the masked data via a five-nearest-neighbor algorithm that includes both floating point operations and rearrangement, and then a communication operation that transfers the data between concurrent sets of MPI tasks in the different components. So there are four distinct MPI operations in a single ping-pong. There are 4.5 million different links (weights) between the T799 grid points and the ORCA025 grid points and 3 million weights for the mapping in the other direction."

P9L13-14 "above 8000 cores per component, the timing is degraded relative to lower core counts. At higher core counts, the timing depends heavily on the MPI performance."
Why do you not see this behaviour in the IS-ENES2 coupling technology benchmark? Is this due to the different grids used in both test cases?

This paper does not mention nor include an analysis or comparison to the IS-ENES2 benchmark.  Having said that, the comment is interesting, and we are currently looking at the benchmark results in the context of these timing tests to better understand the timing differences.  The curves in the IS-ENES benchmark show roughly the same behavior although the absolute timing is quite different.  These differences are likely related to the different resolutions, different mapping files, and different machines used in the two cases.

P10L25 "while for the src+bfb case, the single operation performs slightly worse"
Are you sure that the measurements (10.56 vs 11.89), this statement is referring to, are correct? (5.95 + 6.02) > 10.56 (taken from Table 2 and Table 3)

This is a good point that we should clarify.  The pipo time is done without any barriers while the mapping timing is done as a separate test run with barriers around the mapping.  In general, those barriers will slow the model down because any overlap in mapping and data transfer due to load imbalance will be lost with the barriers.  Timing parallel kernels in a consistent way is always tricky.  We have updated section 3.4, combined tables 2 and 3, and added some new timing information for the pipo time when barriered.  We hope this significantly clarifies the timing information.

P11L13-14 "It's likely that the MPI memory footprint is accounting for most of this behavior (Balaji et al, 2008, Gropp, 2009)."
With a modern MPI implementation this should not happen. I have not seen this behaviour in similar measurements for the ICON model. You could verify this using for example the valgrind tool Massif.

We have updated this sentence to "It is possible that the MPI memory footprint is accounting for most of this behavior (Balaji et al, 2008, Gropp, 2009), but further investigation will need to be carried out in the future to confirm." We hope that is a reasonable response.

P20 Figure 4
In this case, I would not use a trendline or and any line between the measurement points. The number of cores has a significant impact on the decomposition, which might lead to interesting result between the provided measurements. Therefore, a line between the points implies a continuity that might not reflect reality for this test case.

We have removed the line between the measurement points in Figure 4.

P21 Figure 5
Are these single measurements or averages?

We have added 1 sentence in section 3.3 to answer this question, "Two trials were carried out and the results shown are for the best times with variability generally much less than 5% between runs."

P22 Table 2 and 3
(0.69 + 0.60) == 1.29 => Did the data exchange between the components only take a negligible amount of time?

We have rerun the tests with additional timing information, combined tables 2 and 3, updated the table with some additional results, and updated section 3.4 to clarify these results.  The barriered pipo time is now shown to compare with the sum of the map time for an apples and apples comparison.  Compared to the unbarriered pipo time, this also better demonstrates the amount of load imbalance and overlapping work between the mapping and communication in the unbarriered case and the text has been revised to reflect that.

P22 Table 2 and 3
(5.95 + 6.02) > 10.56 => Are the measurements correct?

See comment above.

P22 Table 2 and 3
(11.89 – (4.70 + 4.60)) > (12.15 – (4.86 + 4.97)) => time for mapping ↑ time for transfer ↓ => How do you explain this?

Again, this comes down to the barrier around map timing which we now describe in the text.  See the comment above with regard to P22, Table 2 and 3.  We have added some text in section 3.4 to explain the timing numbers better.  The old timing information did not provide insight into the load imbalance.  In fact, the mapping time does go up but you cannot immediately assume the communication time is decreased.  This is hopefully clarified in the text.

P22 Table 4
(2.11 – 1.29) >> (2.17 – 1.61)
I would assume that the cost for CONSERV is independent of the src/dst option. How do you explain the difference?

It's not clear that you can make simple conclusions like this from the timing information.  The timing of the pipo is complicated by load imbalance, dependencies

in the communication between tasks, and other issues.  In addition, the order of operations for src+bfb and dst+bfb are quite different and depending where in the sequencing the global sums are carried out, this can have an impact on the load imbalance and overall pipo time.   We have updated Table 4 to reflect some new results and we have added some additional information in the discussion in Section 3.5.

**Technical Comments**

P1L14 "separate hub coupler **process**"

We have implemented this change to the text.

P1L23 "OASIS is **a** coupling software"

We have not made this change, we feel the current wording is ok.

P1L32-33 "OASIS-MCT supports coupling of fields on relatively arbitrary grids [...]." Is "OASIS-MCT supports coupling of fields defined on most grid types, commonly used in climate science, [...]." better?

We have updated this sentence consistent with the review.

P1L33 "via a put/get approach. This approach means components make subroutine calls [...]" "via a put/get approach, which is based on components making subroutine calls [...]"?

We have updated this sentence as suggested by the reviewer.

P2L20-21 "calculation of the **source neighbour** weights and addresses needed for the mapping"

We have updated the spelling of neighbor

P2L23-26 check use of Oxford comma

We have added a comma as suggested

P2L25 Why did you use the long form for AWI while using the abbreviation for ECMWF, KNMI, and MPI-M?

We put abbreviations everywhere.

P2L26-28 Maybe add a reference?

We added Hollingsworth et al., 2008

P2L29-30 "OASIS3-MCT extended the widely used and distributed OASIS3 version of the model." "It extends the widely used and distributed OASIS3."?

We have updated the sentence as suggested by the reviewer

P3L8 "Transform**ation**s are carried out" P3L21 "section 4 provides a summary" Section 4 is called "Conclusion"

We have updated this sentence as follows, "... and section 4 provides conclusions and a summary."

P6L2-3 "In OASIS3-MCT, this operation **can is** now **be** performed in parallel on the source or destination processes"
If the bfb option is used, it will still be done in serial, or not?

It will always be done in parallel.  Even if the bfb option is used to compute the global sums, the corrections are applied in parallel on the decomposed fields after broadcasting those global sums to all tasks.  We have added a word, "decomposed" to "...applies corrections to the decomposed mapped fields..." to make it clearer the correction is happening in parallel.

P7L1-2 "are indicated by the different lettered arrows **in Figure 1**."

I assume the reviewer was asking us to check the capitalization of "F"?  We have corrected this throughout the paper and changed all references to figures and tables to small letters unless figure or table are the first word of a sentence.

P9L10 "**there are**"

We have removed "there are" in that compound sentence.

P10L5 "(1.91**s** vs 4.70**s**)"

Units have been added

P10L32 "CONSERV unset"
In Table 4 this is called off.

We have modified table 4 and used the word unset consistently.

P10L31-33 "Table 4 shows [...]. Table 4 shows [...]" Identical start of two consecutive sentences.

We have updated the second sentence starting with "Table 4 shows" to improve readiblity.

P11L4-5 "such as area overlap conservative" Maybe place a reference to:
http://dx.doi.org/10.1175/1520-0493(1999)127%3C2204:FASOCR%3E2.0.CO;2

We have added an equivalent reference as requested by the reviewer

P12L19 "**10stens** of thousands"

We have changed the wording from 10s to tens as suggested

P12L27-28 "**fastestbest** performance"

We have changed the wording of fastest to best as suggested

P16L5-7 "Valcke [...] 2012a"

We have removed this reference and changed 2012b to 2012.

P16L12-14 "Valcke [...] 2015"
Could not find references of these papers in the text.

We have added this reference in Section 1 near the end of the section.

P19 Figure 3
P20 Figure 4
P21 Figure 5
x-axis: maybe use logarithmic base 2 instead of 10 y-axis label: "**secondsTime in s**"

We have renamed the y-axis label, but left the x-axis scale as is.

P19 Figure 3
y-axis: use logarithmic scale to better show behaviour for 1 to 1000 cores per component

We believe the key to this figure is not the time at the lower core counts, but the time at the higher core counts.  Switching the y-axis to log will make that information less clear.  We have not changed the y-axis scale.

P21 Figure 5
The data set "T799->025,dst" seems to have two data points at 24 core per component while all others only have one.

Thanks for catching that, we have corrected that problem by eliminating a redundant point.

P21 Figure 5
For higher number of cores (> 40), the choice of the symbols for the individual data sets makes it hard to read.

This is a good point.  We have changed the symbols and updated the symbol table to make the data more readable.  None of the symbols are filled anymore.

P22 Table 4 "**pescores**" P22L23 "**taskscores**"

We have changed both the pes and tasks wording to cores as suggested.

P23 Figure 6
x-axis: maybe use logarithmic base 2 instead of 10

We have not changed the x or y axis scales.

P23 Figure 6
MB or MiB? per core?

MB is typically used when discussing  memory use.  I don't think it adds to the paper to differentiate between MB and MiB.  They differ in definition by less than 5% and that difference has no impact on the plot or discussions.  In fact, the scaling of the memory use is more important than the absolute memory use numbers in the plot.

**Questions not necessarily relevant for the paper**
P1L19-20 "10,000 two dimensional coupling fields"
In case of 3d fields, would the different levels be counted as separate fields?

In the underlying implementation of the new "bundle" feature, the 3d fields are treated under the covers are multiple 2d fields.  We count multi-level 3d fields as multiple 2d fields.  The requirement for using 2d bundled field is the same as the requirement for coupling multiple fields in a single namcouple statement, i.e. those fields have to share the same grids, masks and will use the same mapping file.

P7L24 "OASIS3-MCT provides some new capabilities to detect potential deadlocks before they occur" Very interesting! Can you be more specific?

Several checks were added like making sure time didn't go backwards, making sure a coupling period wasn't skipped, and others.  Some of the new checks had to be removed or deprecated to support sequential coupling on overlapping pes.  In general, the new capabilities are not adequate to prevent deadlocks.

P7L28-29 "In OASIS3-MCT, puts are always non-blocking while gets are blocking."
Are there plans for non-blocking gets?

There are no plans for non-blocking gets. In general, we presume that users execute a get when the data is needed. A non-blocking get would require users add a wait in their code before they could use the data which we think adds complexity with little gain. There is lack of symmetry with respect to put and get in systems such as this. If the community requests non-blocking gets, they could probably be implemented but with some additional burden on users and the user implementation.

P12L11-12 "the cost associated with generating the mapping files can be moved to a preprocessing step" which not necessarily has to be faster, if weight computation is done in parallel.

This is true. But right now, Oasis3-MCT does not provide an on-line parallel weights computation capability. Several offline tools do provide that capability. In addition, those offline tools have experts in grid and weights generation that cannot (and maybe should not) be duplicated within Oasis. The complexity associated with generating weights on (for instance) complex unstructured grids, and for many different types of gridding options (bilinear, conservative, higher order, gradient preserving, nearest neighbor, and so forth) are probably best dealt with by specialized tools outside Oasis, and these tools do already exist and exceed any capability that Oasis could build. Having said that, if future requirements, such as time evolving grids impose new requirements on Oasis for fast, parallel weights generation, Oasis will consider incorporating additional external tools into the infrastructure. This section of text in the paper was updated to reflect these ideas.

---

## Author Comment (AC3)

Reply to Anonymous Referee #3.

We would like to thank referee #3 for taking the time to review our paper. Our replies to the three main comments are below (in green).

1) There is no discussion of OpenMP as an alternative to MPI. Future hardware will require going to more shared memory and less message passing.

We acknowledge that there is no discussion of OpenMP in the current paper. OpenMP parallelization is not currently explicitly supported in Oasis3-MCT, and OASIS developers are aware of this shortcoming in the current implementation, especially in regards to possible future architectures. As indicated in the final section of the paper, OpenMP parallelization and performance of Oasis3-MCT on new architectures is something that is currently being explored by the development team. We hope to provide support in the next year (or so) and will have results at that time to share with the community.

2) There is no discussion of GPUs, MICs, etc and plans to port OASIS to novel architectures.

The OASIS development team is actively pursuing access and testing of OASIS on newer architectures and hope to have some results in the next year to share with the community. We recognize this is an important issue moving forward.

3) I am somewhat taken aback by the extreme cost of providing bfb (bit for bit) reproducing algorithms. In other similar codes this cost ratio is somewhat lower (which could of course mean that the non-reproducing modes in other codes are too slow!) This may require some work.

With regard to the cost of the bfb conservation computation, we were also quite shocked at the cost of the bfb operation. We have revised the discussion and results of the CONSERV transform in the paper significantly, adding global sum options that have been recently added to the OASIS infrastructure and that will be released in OASIS3-MCT_4.0. The OASIS3-MCT_3.0 timings showed a clear problem with the bfb CONSERV performance. OASIS3-MCT_4.0 will provide additional options, including an option called "reprosum" that produces bit-for-bit results on different core counts and decompositions while performing significantly better than the current "bfb" option. Please see revised section 2.4, 3.5, and table 3 in the paper.